

# Molecular characterization of G-protein-coupled receptor (GPCR) and protein kinase A (PKA) cDNA in *Perinereis aibuhitensis* and expression during benzo(a)pyrene exposure

Yi Huang[1,*], Jia Sun[1,*], Ping Han[1], Heling Zhao[2], Mengting Wang[1], Yibing Zhou[1], Dazuo Yang[1] and Huan Zhao[1]

[1] Key Laboratory of Marine Bio-Resources Restoration and Habitat Reparation in Liaoning Province, Dalian Ocean University, Dalian, Liaoning, China
[2] Asian Herpetological Research Editorial Office, Chengdu Institute of Biology, Chinese Academy of Sciences, Chengdu, Sichuan, China
* These authors contributed equally to this work.

Corresponding author
Huan Zhao, zhaohuan@dlou.edu.cn

## ABSTRACT

**Background:** G-protein-coupled receptors (GPCRs) are one of the most important molecules that transfer signals across the plasma membrane, and play central roles in physiological systems. The molecular architecture of GPCRs allows them to bind to diverse chemicals, including environmental contaminants.
**Methods:** To investigate the effects of benzo(a)pyrene (B(a)P) on GPCR signaling, GPCR and the protein kinase A (PKA) catalytic subunit of *Perinereis aibuhitensis* were cloned. The expression patterns of these two genes during B(a)P exposure were determined with real-time fluorescence quantitative PCR. The PKA content in *P. aibuhitensis* under B(a)P exposure was examined.
**Results:** The full-length cDNAs of *PaGPCR* and the *PaPKA* catalytic subunit were 1,514 and 2,662 nucleotides, respectively, encoding 338 and 350 amino acids, respectively. Multiple sequence alignments indicated that the deduced amino acid sequence of PaGPCR shared a low level of similarity with the orphan GPCRs of polychaetes and echinoderms, whereas PaPKA shared a high level of identify with the PKA catalytic subunits of other invertebrates. B(a)P exposure time-dependently elevated the expression of *PaGPCR* and *PaPKA*. The expression of both *PaGPCR* and *PaPKA* was also dose-dependent, except at a dose of 10 µg/L B(a)P. The PKA content in concentration group was elevated on day 4, with time prolonging the PKA content was down-regulated to control level.
**Discussion:** These results suggested that GPCR signaling in *P. aibuhitensis* was involved in the polychaete's response to environmental contaminants.

## INTRODUCTION

Benzo(a)pyrene (B(a)P), a kind of polycyclic aromatic hydrocarbon (PAH), can cause genetic damage, immune and endocrine dysfunction, and malformation in humans and
other organisms. Its high lipophilicity allows it to absorb to organic matter and other particulate matter and thus accumulate in sediments. Recent increases in offshore oil production and transportation and the sewerage discharge of domestic and industrial wastewater have led to environmental pollution in coastal regions, and B(a)P has been widely detected in sediments around the world, even in China. The levels of B(a)P in the sediments of Dalian Bay vary from 10.5 to 3421.2 ng/g (*Zhang, 2008*), and in the sediments around the drilling platform in the Bohai Sea, the concentration of B(a)P is up to 27.69 ng/g (*Yang et al., 2016*). PAH such as B(a)P can be absorbed by benthic organisms via ingestion or through their body surfaces, and B(a)P is reported to have serious effects on deposit feeders. Therefore, the toxicity and bioavailability of B(a)P are important factors in the assessment of sediment pollution.

G-protein-coupled receptors (GPCRs) are the largest superfamily of cell membrane proteins (*Fredriksson et al., 2003*). The molecular architecture of the GPCRs allows them to bind to diverse organic and inorganic molecules. GPCRs mediate cell proliferation and survival by transmitting signals from a range of extracellular ligands across the cell membrane to signaling pathways. In vertebrates, they are key regulators of the innate and adaptive immune responses and have been investigated as potential targets in drug discovery (*Garland, 2013*). However, examples of GPCRs in invertebrates are limited. *Miller et al. (2015)* reported that *Caenorhabditis elegans* with mutations in the GPCR follicle-stimulating hormone receptor 1 (FSHR-1) died significantly more quickly in the presence of cadmium than wild-type nematodes, which suggests that this GPCR pathway protects the nematode against cadmium-induced damage. They also found that FSHR-1 antagonizes the capacity of *C. elegans* to resist cold stress, and the mutants lacking *fshr-1* survived better than wild-type worms at low temperatures. *Dong & Zhang (2012)* identified a putative GPCR gene, *HP1R*, in the red swamp crayfish *Procambarus clarkia*, and the expression of *HP1R* was significantly increased in the presence of Gram-negative bacteria.

Because the aromatic structures present a number of GPCR ligands, GPCRs are potential targets of aromatic pollutants such as B(a)P (*Le Ferrec & Øvrevik, 2018*). *Mayati et al. (2012)* reported the interaction between B(a)P and the $\beta_2$-adrenergic receptor ($\beta_2$ADR) in endothelial HMEC-1 cells and the consequent increase in intracellular $Ca^{2+}$, which influenced the expression of cytochrome P450 B1. This suggests that $\beta_2$ADR, a kind of GPCRs, is potentially involved in the deleterious effects of B(a)P. *Factor et al. (2011)* also observed the reduced expression and function of $\beta_2$ADR in airway epithelial cells and smooth muscle cells after their exposure to a mixture of PAHs. This implies that the $\beta_2$ADR signal transduction pathway is affected by PAHs. These data indicate that PAHs, including B(a)P, modulate the concentrations of intracytosolic cyclic adenosine monophosphate (cAMP) or $Ca^{2+}$ via G-protein-dependent mechanisms (*Bainy, 2007*; *Nadal et al., 2000*).

The marine polychaete *Perinereis aibuhitensis* is widely distributed in the mudflats and estuarine sediments that occur widely along the coasts of Southeast Asia. They spend most of their lives within the sediments, ensuring their continuous contact with any sediment-associated contaminants. *Chen et al. (2012)* identified a *CYP4* gene of

*P. aibuhitensis* and showed that exposure to petroleum hydrocarbons significantly induced the expression of this gene. To clarify whether GPCR signal transduction pathway was involved in modulating the toxicity of aromatic pollutants, the full-length *GPCR* and protein kinase A (*PKA*) cDNAs were cloned and the expression patterns of these two genes were determined in this study. Our results provide important information on the function of GPCRs in polychaetes.

## MATERIALS AND METHODS

### B(a)P exposure

*Perinereis aibuhitensis* specimens (10–15 cm, 2.0 ± 0.5 g wet weight) were collected from Dalian Dongyuan aquaculture farm at the estuary of Jinzhou Bay in Dalian, China. We have a long-term cooperation agreement with the farm. The agreement permits us to collect research samples from all their aquaculture sites including a certain estuary area under their ownership (The field permit is attached in the Supplemental Files). The animals were transferred to the laboratory and acclimatized in filtered seawater (salinity 31–32, temperature 16 ± 0.5 °C) for a week before the experiment. During acclimatization, the *P. aibuhitensis* were fed a powdered mix containing kelp powder, gulf-weed powder, fish meal, yeast, and spirulina powder. The worms were deprived of food during their exposure to B(a)P.

Based on the standard seawater quality of the People's Republic of China (GB 3097-1997), four B(a)P concentration groups were established: 0.5, 5, 10, and 50 μg/L. A blank (seawater only) group and an acetone control group (100 μL/L) were also established. Three repetitions of each concentration group were set up. Ten worms were randomly placed in 2L beakers containing different concentrations of B(a)P. During the experiment, the temperature of the seawater was 16 ± 0.5 °C, and the seawater was renewed every 24 h. On days 4, 7 and 14 of the experiment, four individuals were randomly sampled from each concentration group, and the body wall was removed for gene expression analysis. Three individuals was randomly sampled for PKA content analysis.

### Cloning the full-length *GPCR* and *PKA* cDNAs of *P. aibuhitensis*

Three worms in blank group were ground to powder and the total RNA was extracted with RNAiso™ Plus (TaKaRa, Dalian, China). The quality of the RNA was determined with 1% agarose gel electrophoresis. The RNA (500 ng) was reverse transcribed to cDNA for the rapid amplification of cDNA ends (RACE) using the SMARTer® RACE Kit (Clontech, Palo Alto, CA, USA). The 3′ and 5′RACE primers were designed with the Primer 5.0 software (PREMIER Biosoft, Palo Alto, CA, USA) according to the confirmed partial sequences of *GPCR* and *PKA* obtained from *P. aibuhitensis* transcriptome sequences in our laboratory (unpublished). The primers used in this study are shown in Table 1.

The 3′RACE amplification of *P. aibuhitensis GPCR* (*PaGPCR*) was performed using the 3′RACE cDNA as the template. The PCR system (50 μL) for *PaGPCR* contained 15.5 μL of PCR-grade water, 25.0 μL of 2× SeqAmp Buffer, 1.0 μL of SeqAmp DNA polymerase, 2.5 μL of 3′RACE cDNA, 5.0 μL 10 × UPM (universal primer mixture), and 1.0 μL of

 

**Table 1 The primers used in this study.**

| Primer name | | Sequence (5′–3′) |
|---|---|---|
| RACE | GPCR–F1 | TGAGAAACGTCGAAGCGAAAGG |
| | GPCR–R1 | ATATTCGACGCTGACCCTAAGGGPCR-R2 GAAACACAGAAGCCACCAGGTC |
| | PKA-F1 | GGATACCCACCTTTCTTTGCTGACC |
| | PKA-F2 | GGTGCGCTTCCCATCTCACTTT |
| | PKA-R1 | CAATAGCGCAGCCTCAGGGACA |
| | UPM Long | CTAATACGACTCACTATAGGGCAAGCAGTGGTATCAACGCAGAGT |
| | UPM Short | CTAATACGACTCACTATAGGGC |
| Real time PCR | β-actin-R | CGAAGTCCAGAGCAACATAG |
| | β-actin-F GPCR-R3 | CGAAGTCCAGAGCAACATAGCA CCGTAAAAGCCTCATCAAGACA |
| | GPCR-F3 | TTGGCAGGTGTAAATGAATGG |
| | PKA-F3 | GACCAGCCAATCCAAATCTATG |
| | PKA-R3 | GACCCCATTCTTCAGGTTTCC |

primer GPCR-F1 (10 µM). The thermal cycling conditions were: 35 cycles of denaturation at 94 °C for 30 s, annealing at 65 °C for 30 s, and extension at 72 °C for 3 min. The 3′RACE amplification of *P. aibuhitensis PKA* (*PaPKA*) was performed with nested PCR. The outer PCR reaction system for *PaPKA* was the same as that for *PaGPCR*, except that a specific primer was used. The reaction conditions for the outer PCR were: 35 cycles of denaturation at 94 °C for 30 s, annealing at 63.1 °C for 30 s, and extension at 72 °C for 3 min. The outer PCR product (5.0 µL) was diluted with 245 µL of TE buffer, and 5.0 µL of the diluted product was used as the template for the inner PCR. The reaction conditions and system for the inner PCR were the same as for the outer PCR of *PaPKA*.

The 5′RACE product of *PaGPCR* was amplified with nested PCR. The outer PCR reaction system (50 µL) for *PaGPCR* contained 15.5 µL of PCR-grade water, 25.0 µL of 2× SeqAmp Buffer, 1.0 µL of SeqAmp DNA polymerase, 2.5 µL of 5′RACE cDNA, 5.0 µL of 10 × UPM, and 1.0 µL of primer GPCR-R1 (10 µM). The reaction conditions for the outer PCR were: 35 cycles of denaturation at 94 °C for 30 s, annealing at 60 °C for 30 s, and extension at 72 °C for 3 min. The outer PCR product (5.0 µL) of *PaGPCR* was diluted with 245 µL of TE buffer and 5.0 µL of the diluted product was used as the template for the inner PCR. The reaction system (50 µL) for the inner PCR of *PaGPCR* contained 5.0 µL of the diluted outer PCR product, 17.0 µL of PCR-grade water, 25.0 µL of 2 × SeqAmp Buffer, 1.0 µL of SeqAmp DNA Polymerase, 1.0 µL of UPM Short, and 1.0 µL of primer GPCR-R2 (10 µM). The reaction conditions were: 20 cycles of denaturation at 94 °C for 30 s, annealing at 60 °C for 30 s, and extension at 72 °C for 3 min. The 5′RACE of *PaPKA* was amplified with ordinary PCR, and the reaction system and conditions were the same as those for *PaGPCR*.

The PCR products were detected with 1% agarose gel electrophoresis and purified with the Agarose Gel DNA Purification Kit (Tiangen, Beijing, China), according to the manufacturer's instructions. The PCR products were sequenced by Takara Biotechnology Co. Ltd.

## Bioinformatic analysis of PaGPCR and PaPKA

The amino acid sequences of *PaGPCR* and *PaPKA* were deduced with the Expert Protein Analysis System (http://www.us.expasy.org/tools). The conserved domain in each amino acid sequence was analyzed with the Motif Scan (https://myhits.isb-sib.ch/cgi-bin/motif_scan) and Expasy (https://prosite.expasy.org/). The protein localization sites in the cell were predicted with the Psort software (http://psort.hgc.jp/form2.html). The transmembrane (TM) helix in the protein were predicted with the TMHMM software (http://www.cbs.dtu.dk/services/TMHMM/). The tertiary structures of PaGPCR and PaPKA were predicted with the Swiss-Model software (http://swissmodel.expasy.org/interactive). Multiple sequences were aligned with the Clustal W software (https://www.ebi.ac.uk/Tools/msa/clustalw2/). Phylogenetic analysis of GPCR and PKA were performed in MEGA 5.0. The tree topologies were evaluated with bootstrapping, using 1,000 replicates.

## Expression of *PaGPCR* and *PaPKA* genes during B(a)P exposure

Real-time fluorescence quantitative PCR was used to investigate the expression of the two genes in *P. aibuhitensis* during B(a)P exposure. The β-actin gene was used as the reference gene, according to our previous study (*Li et al., 2018*). The primer information is shown in Table 1. Amplification was performed in 20 µL reaction system containing 10 µL of SYBR Premix Ex Taq II (Tli RNaseH Plus)(TaKaRa, Dalian, China), 0.8 µL of each primer (10 µM), 0.4 µL of 50× ROX Reference Dye II, 2.0 µL of cDNA, and 6.0 µL of $H_2O$. The reaction conditions were: 95 °C for 30 s, then 40 cycles of 95 °C for 5 s and 60 °C for 34 s. The melting curves were analyzed after the real-time quantitative PCR. The standard curves were tested with serial 10-fold sample dilutions. The slopes of the standard curves and the PCR efficiency were calculated to confirm the accuracy of the real-time PCR data.

## PKA content in *P. aibuhitensis* under B(a)P exposure

Body wall (about 100 mg) of each sample was homogenized in 0.9 mL cold phosphate buffer saline with pH 7.4. The homogenate was centrifuged at 4 °C at 3,000 rpm/min for 15 min. The supernatants were assayed for PKA content using the non-radioactive PKA assay kit (Kexing, Shanghai, China) with the method of ELISA according to manufacturer's protocol. Results are expressed as ng/mL.

## Statistical analysis

The relative quantitative ($2^{-\Delta\Delta Ct}$) method was used to analyze the expression of the *PaGPCR* and *PaPKA* genes. The data are expressed as means ± standard deviations (SD), and one-way analysis of variance was used to analyze the significance of the differences among the different concentration groups at each sampling point, with the SPSS 19.0 software. *P* values ≤ 0.05 were considered statistically significant.

# RESULTS

## Molecular characterization of *PaGPCR*

The 5′RACE and 3′RACE products of *PaGPCR* was 1082 and 800 bp, respectively (see the Supplemental File of the PCR database), and the full-length cDNA of *PaGPCR* was

```
  1 TAAACAGTGGCATTCACGCAGAGTACATGAGGGCAGTTTAAGTGATTCATTTTATGGGAATGGGTTTCCCTGGTG
 76 ATTAGTTAGCAAAAATATATATTTCTGGAATATTTGGACTTCAGAAATTGAAGGATTTTATACTTGTTCTTCGGC
151 TTAATGGAAATTTCATCTTGTGATTGAGGATTTAATCGGTGAATTTTATTACATGTTAACATTATGGATAACACA
  1                                                                     M  D  N  T
226 ACATTCAACAGAACGTTTGATGGGAGTTTGAACCCTAACTTCAACTACATTGGAGATTTTGTGGTGTACATAGTA
  5  T  F  N  R  T  F  D  G  S  L  N  P  N  F  N  Y  I  G  D  F  V  V  Y  I  V
301 ATCGGATGTCTTGGGATTTTAGATAATGGATTTGTTATTATAGTCATTCTCCATAGCAGAAAGATGAGGAATAAA
 30  I  G  C  L  G  I  L  D  N  G  F  V  I  I  V  I  L  H  S  R  K  M  R  N  K
376 CTGTGCAATTTATTCATCCTTAATCAAAGTGTGGTAGACCTGGTGGCTTCTGTGTTTCTCCTGTGCAATTCTCCG
 55  L  C  N  L  F  I  L  N  Q  S  V  V  D  L  V  A  S  V  F  L  L  C  N  S  P
451 TCTGTTCCGACCTTAGGGTCAGCGTCGAATATTAGTCTGGAGTTTTATTGCCGCATTTGGGATTCGAACTATCTC
 80  S  V  P  T  L  G  S  A  S  N  I  S  L  E  F  Y  C  R  I  W  D  S  N  Y  L
526 TTCTGGGCTGCCGTCACATGGTCAACTTACAACTTAGTCGCCATCACAATCGAACGTTACTTAGAGGTCGTTCAC
105  F  W  A  A  V  T  W  S  T  Y  N  L  V  A  I  T  I  E  R  Y  L  E  V  V  H
601 CCACTTCGGTACAGATCATTCTTCACGCGGAGACGTGCAAAGGTCATTGTCGCTGTCGTCTGGTTGGTTGGATTC
130  P  L  R  Y  R  S  F  F  T  R  R  R  A  K  V  I  V  A  V  V  W  L  V  G  F
676 ACCATACCTATCGTGACGTCAGTTATCACCAGTCCTGCGGGAGCAGACGGCACTTGTCAGAAGCACAGAGCGTGG
155  T  I  P  I  V  T  S  V  I  T  S  P  A  G  A  D  G  T  C  Q  K  H  R  A  W
751 TCCTCCCGACTCATGGCTGCCCTCGTAGGATTTTACGCCCTCTTCTTCGGATTTCTTTTACCTGTCGTCATAATG
180  S  S  R  L  M  A  A  L  V  G  F  Y  A  L  F  F  G  F  L  L  P  V  V  I  M
826 ATCGTTTGCTACACTCAGATGATCATGACCTTCAACTTGAAGGTCCGACCCTCCGACCCCAGCACAATGATCTCC
205  I  V  C  Y  T  Q  M  I  M  T  F  N  L  K  V  R  P  S  D  P  S  T  M  I  S
901 GAAAGTGAGAAACGTCGAAGCGAAAGGATGTTGAGGGTCCGTAAAAGCCTCATCAAGACAATGTTGATGGTTTCT
230  E  S  E  K  R  R  S  E  R  M  L  R  V  R  K  S  L  I  K  T  M  L  M  V  S
976 ATCGTCTTTGTGATCTGTTGGATCGGCGACCAAGTTTATTTCTTCCTCTTCAACATCAGAGTCATCAAAGACCTT
255  I  V  F  V  I  C  W  I  G  D  Q  V  Y  F  F  L  F  N  I  R  V  I  K  D  L
1051 CAACAAACTCTTACGACTATCGTTGTTTCGTTAGCTTTCCTCAATTGCTGCATTAATCCATTCATTTACACCTGC
280  Q  Q  T  L  T  T  I  V  V  S  L  A  F  L  N  C  C  I  N  F  F  Y  T  C
1126 CAATATAACGACTTCCAAGAAGCTACAAGAAGACTCTTAAAAATCAAAAAGGAGAGTGAAAACAGTGAAAGGTCT
305  Q  Y  N  D  F  Q  E  A  T  R  R  L  L  K  I  K  K  E  S  E  N  S  E  R  S
1201 ACGTTGGATCTGTCTAACCAAAAAGTCTAATCCACATGAACTAATTAAAACATATTGATTCGACCAATTTTTAAC
330  T  L  D  L  S  N  Q  K  V  *
1276 TTTCTTACAGATTTACCAAAATTAATTTTTAATTTTAGTTACCTCTATTTATTTAATATTTTCATGTACAGCGCC
1351 TCTGAAGATTTTACTTTCCGAGATCTCGCTACATAAATAAGTCATTATTTGTATTTATTTTTAAAGAGGTAATGA
1426 CAAATTGTAAAGATGTTCTTAATTTGTGATGGTCTCCATTAAATTTGACATGGTCATTTACTAAAAAAAAAAAAA
1501 AAAAAAAAAAAAA
```

**Figure 1 Nucleotide sequence and deduced amino acid sequence of GPCR from *Perinereis aibuhitensis*.** Initiation codon (ATG) and termination codon (TAA) are highlighted in red boxes. The seven-transmembrane (7TM) domains (TM I to TM VII) are underlined with red lines. The E/DRY and NPXXY motifs are in shadow.

obtained by sequence assembly. The full-length cDNA of *PaGPCR* was 1,514 bp and included a 5′ untranslated region (UTR) of 213 bp, a 3′UTR of 284 bp, and an open reading frame (ORF) of 1,017 bp, encoding 338 amino acids with a predicted molecular weight of 38.799 kDa and a theoretical isoelectric point of 9.38 (Fig. 1). This nucleotide sequence was deposited in the GenBank database under accession number KX792261.

The seven-transmembrane (7TM)-helix bundle (304–1,146 bp) that defines the GPCR protein family was present in *PaGPCR*. The glutamic acid/aspartic acid-arginine-tyrosine (E/DRY) motif (amino acids 122–124) at the border between TM III and intracellular loop 2 and the NPXXY motif (amino acids 298–302) of TM VII near the inner cell membrane were detected in the deduced protein sequence, indicating that the protein
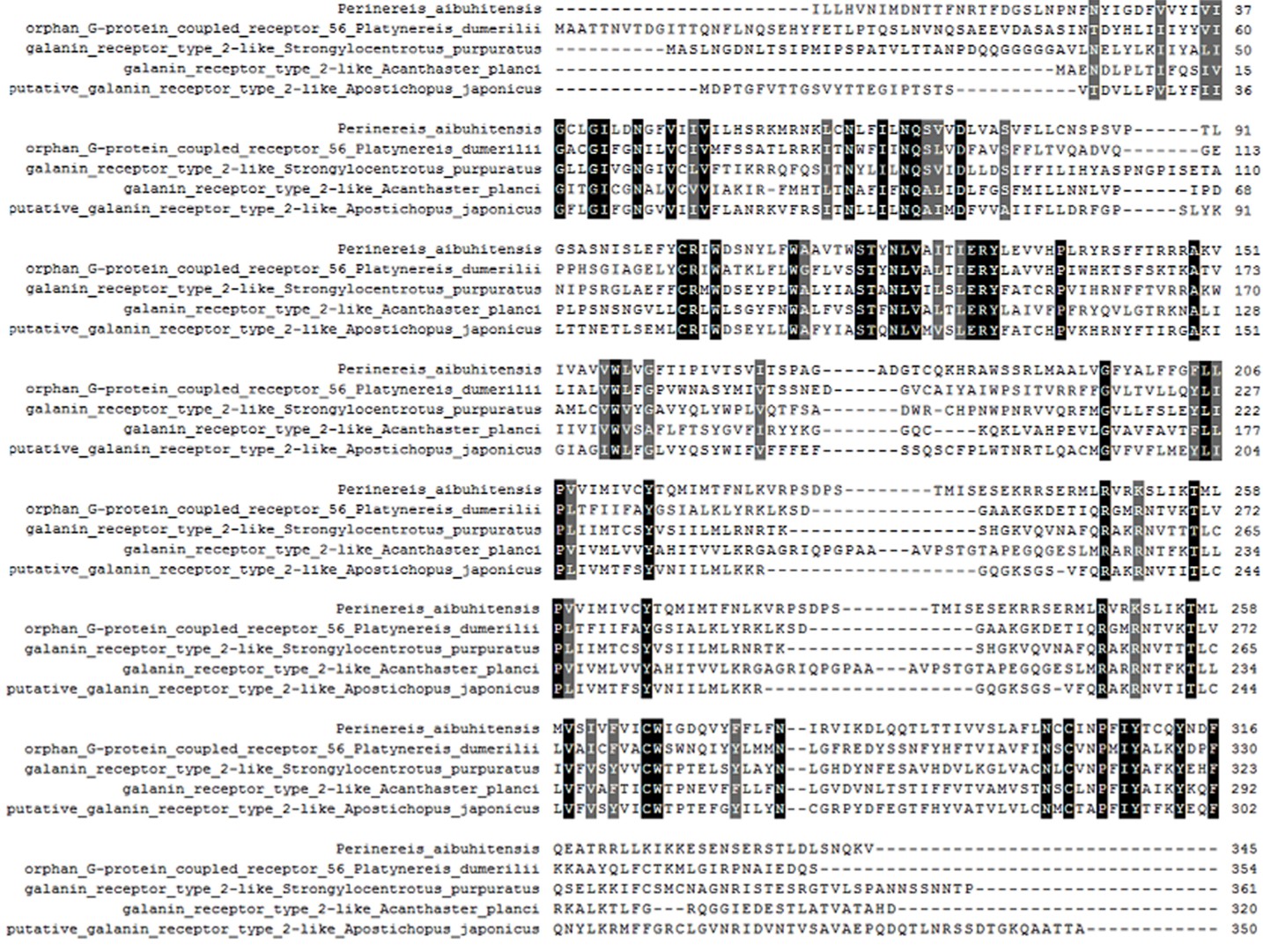

**Figure 2 Multiple alignment analysis of PaGPCR with other GPCR protein.** Amino acid residues that are conserved in at least of 50% sequence are shaded and similar amino acids are shaded in dark. The GenBank accession number for these proteins are as follows: (*Platynereis dumerilii* orphan G protein coupled receptor, 56AKQ63061.1; *Strongylocentrotus purpuratus* galanin receptor type 2-like, XP_003727596.1; *Acanthaster planci* galanin receptor type 2-like, XP022098630.1; *Apostichopus japonicus* putative galanin receptor type 2-like, PIK48567.1).

sequence belonged to the rhodopsin family. In an amino acid comparison, PaGPCR shared 33% similarity with the orphan GPCR of *Platynereis dumerilii* and 30–33% similarity with galanin receptor type 2 of echinoderms (Fig. 2).

The predicted cellular localization of the PaGPCR protein showed it mostly located on the cell membrane (52.2%), and seven TM helices were predicted in the deduced protein sequence (Fig. 3). The three-dimensional structural analysis of PaGPCR showed that it contained seven α-helices, similar to the GPCRs of other animals (Fig. 4). Its three-dimensional structure and protein localization confirmed that this protein sequence was a GPCR.

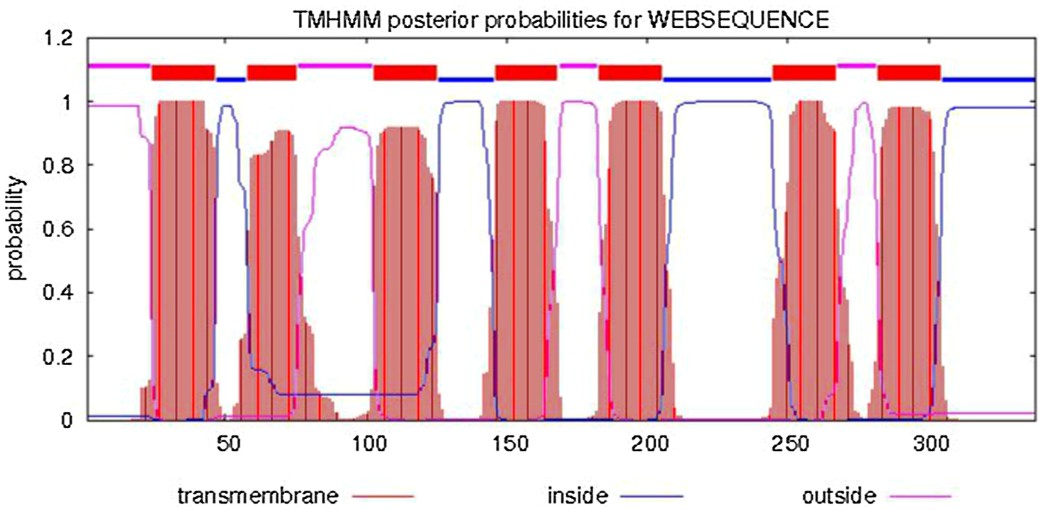

**Figure 3  Analysis of transmembrane region of PaGPCR.** The whole sequence is labeled as inside (blue line) or out side (pink line), and the transmembrane region was labeled with red line.

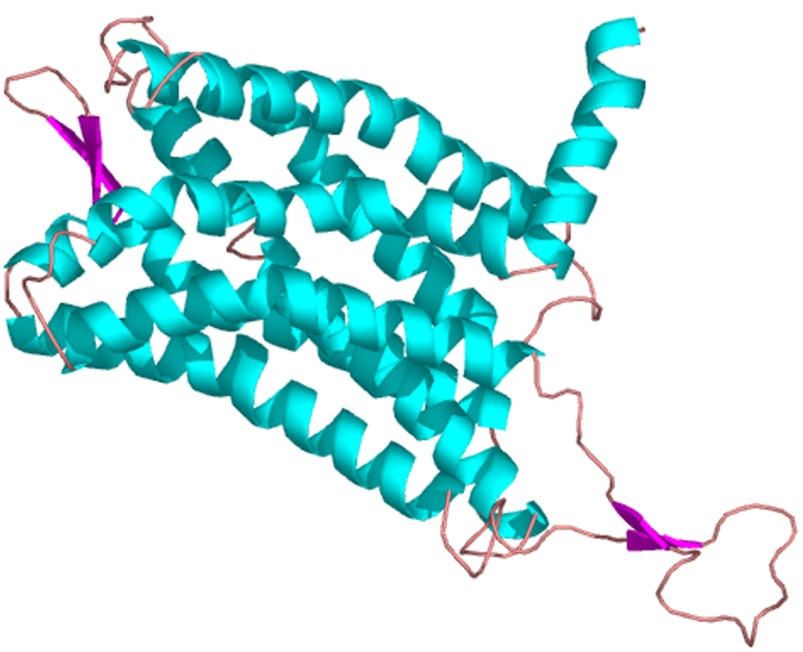

**Figure 4  The three dimensional structure of PaGPCR.** The helix is colored by blue, the sheet is colored by magenta, and the loop is colored by salmon.

## Molecular characterization of *PaPKA*

The 5′RACE and 3′RACE products of *PaPKA* was 1918 and 1765 bp, respectively (see the Supplemental File of the PCR database), and the full-length cDNA of *PaPKA* was obtained by sequence assembly. The total length of *PaPKA* cDNA was 2662 bp, containing a 3′UTR of 1,483 bp, a 5′UTR of 126 bp, and an ORF of 1,053 bp encoding 350 amino acids (Fig. 5). The predicted molecular weight of PaPKA was 40.28 kDa and its theoretical

```
   1 GTCGAAGAGATCGAGGTGGAATATTCAGACATAATTTTTGAGAAGCTGGTTCTGAGAGTTCTCTGATTTCTGGCCGGTTAATTCCTCTGGATCACCACGGACTAG
 106 GTAGTTTACCACACGGTAGCCATGGGAAATGCTGCAACAGCAAAGAAAGGCGATCCAGAGAATGTCAAAGAGTTCTTAGCCAAAGCTAAAGAGGACTTCAACAAG
   1                                          M   G   N   A   A   T   A   K   K   G   D   P   E   N   V   K   E   F   L   A   K   A   K   E   D   F   N   K
 211 AAATGGGAAGAGCCATCATGTAACACTGCATCACTAGATGACTTCGACAGAATTAAAACCCTGGGAACAGGGTCATTTGGACGGGTCATGCTGGTTCAGCACAAA
  29  K   W   E   E   P   S   C   N   T   A   S   L   D   D   F   D   R   I   K   T   L   G   T   G   S   F   G   R   V   M   L   V   Q   H   K
 316 GCCACGAAGGAGTACTATGCCATGAAGATTTTAGATAAACAAAAGGTAGTGAAACTGAAGCAAGTTGAACACATTGAATGAAAAGAAAATTCTGTCCGCCATA
  64  A   T   K   E   Y   Y   A   M   K   I   L   D   K   Q   K   V   V   K   L   K   Q   V   E   H   T   L   N   E   K   K   I   L   S   A   I
 421 TCATTTCCATTCTTAGTGAGCCTAGAGTACAGTTTTAAGGATAACTCAAATTTGTACATGGTATTGGAGTTCGTGACAGGAGGTGAAAGTTCTCACATCTGCGA
  99  S   F   P   F   L   V   S   L   E   Y   S   F   K   D   N   S   N   L   Y   M   V   L   E   F   V   T   G   G   E   M   F   S   H   L   R
 526 AGAATTGGCCGATTTAGTGAAACTCACAGCCGATTTTATGCTGCACAAGTATGCATGGTATTTGAATATCTGCACAATCTAGACACACTGTACAGAGATTTGAAG
 134  R   I   G   R   F   S   E   T   H   S   R   F   Y   A   A   Q   V   C   M   V   F   E   Y   L   H   N   L   D   T   L   Y   R   D   L   K
 631 CCAGAAAATATTCTGATTGATGACACTGGTCACTTGAGAGTAACAGACTTCGGTTTTGCCAAACGCGTAAAAGGCAGGACATGGACGTTGTGTGGCACACCAGAG
 169  P   E   N   I   L   I   D   D   T   G   H   L   R   V   T   D   F   G   F   A   K   R   V   K   G   R   T   W   T   L   C   G   T   P   E
 736 TACCTGGCCCCAGAAATCATCTTGAGCAAGGGCTACAACAAAGCCGTAGACTGGTGGGCGCTTGGAGTCCTTGTTTATGAAATGGCAGCTGGATACCCACCTTTC
 204  Y   L   A   P   E   I   I   L   S   K   G   Y   N   K   A   V   D   W   W   A   L   G   V   L   V   Y   E   M   A   A   G   Y   P   P   F
 841 TTTGCTGACCAGCCAATCCAAATCTATGAGAAGATTGTCTCAGGAAAGGTGCGCTTCCCATCTCACTTTAGTTCTGATTTGAAGGATCTTTTGAAGAATCTGCTA
 239  F   A   D   Q   P   I   Q   I   Y   E   K   I   V   S   G   K   V   R   F   P   S   H   F   S   S   D   L   K   D   L   L   K   N   L   L
 946 CAGGTAGACTTGACAAAACGTTATGGAAACCTGAAGAATGGGGTCAACGATATCAAGAATCACAAGTGGTTCTCCACCACAGACTGGATTGCTATCTACCAGAAA
 274  Q   V   D   L   T   K   R   Y   G   N   L   K   N   G   V   N   D   I   K   N   H   K   W   F   S   T   T   D   W   I   A   I   Y   Q   K
1051 AAGGTTGAAGCACCCTTCATTCCCAAGTGCAAAGGCCCAGGTGACTACAGCAACTTTGATGACTATGAGGAAGAACCACTGAGAATTTCGTCAACGGAAAAATGT
 309  K   V   E   A   P   F   I   P   K   C   K   G   P   G   D   Y   S   N   F   D   D   Y   E   E   E   P   L   R   I   S   S   T   E   K   C
1156 GCCAAGGAGTTTGCAGACTTCTGAGACAGTTGGCTTGGTAGGTGGCTAGTGTGTGTTGGGGGAACTAAAGACTGTGATGACCATTCTGTGTTCATGGTCTATCACT
 344  A   K   E   F   A   D   F   *
1261 GGCAGCTGGCTGAAGATCTGCGGCCCACATACCACACAATCTCTCAAAGAATACATCCTTTGTGCATTAACAAATGTATATATACTTAATCATATCACATTGCGT
1366 AATTGTAACAAGGTGAATTCAGTTTCATATGTGTCAATGAGGTAAGAGAGAGGGCTTCGATCCCAGGTGTGGCATGTCCCTGAGGCTGCGCTATTGGTCCGAGGT
1471 AAGCCAGGTCGCTTCAGCTCGCACCCCAGGCTGCCTCAAAAGCTACCTTGGTCGTCTACTCGCCGAACTAGTCATTTACCTTGCTGTGTGGCCTTCCAGATTTGT
1576 GTGGGGCAGACTGGTGGAGGATTAGGCTTTAATTGCTTTGCTGAAGTGCCCCACACGGGTCCAGAAGCCAAGGCTATCTTCAAATACACAGTTAATACCATATAT
1681 TAACTTATCATTACAACCTTACTGACCACTTGTTCACATTTAGGTCATTGTCTGCACTTTTATTGTATACAGGTGGTCCATCTGAAACAATTCTCTGAACATTTG
1786 AATTATATCCATTGTAAGGAAGTGCAAACTTCTCTTGTGCTGTGCTCCTCTCCACTCCCGTGCTGCCTCTTTCTCTGCTGTGAACTTTGAATCAGAGAGAAACAA
1891 CACTTTATTTGTATGATGTCAAATATTGTACTTTTGGTAAGGCCACCTCCTAACTGGGTGGTCATCAAGACTTGTATATAGTTTGTGAGCAGTTGGCTTTTGGGT
1996 AGAGTAGGAGTTTTCTGTTCATCATACTGGTGGGATGACACCATCTCCCGCATACCAGTAGTACTGTGTAGTAGCAATAGCCGGCCATAAGGCCTATGAGTCTCC
2101 ATTGTTTCAGCTAATGTAGCTAGGCTTCAAGTCATAAGTCATGATAACATATAATGGCATTATTTGTAGTACTGATGATGAGAGAATTATCTTTGTCAAGGACA
2206 AACTTATTTTACTTTTATATAACTGACTTGTATGTACAGACGTCAAGGCTGTATGAACTGAGAAATTGTTGCTTGTTATGTGTAGTGTATAAAATGTGTATATCT
2311 ATCTCTGCATATTTTGTGTTTGATTGTCAATCCACAAATGCTTCAAATTTATCTTGAAACCATGTACTTGACTAATCTAATTTGTGATTTGAAAAGGTAATTGGA
2416 GTTGTGCTGAATTAGATAGGTATGCTTCCTGACATTGTGTAACACTACTACAGTGAGTCCCATTTGTCATGGGTCCTACTCATGAATGTTGTTATATTTTGTATA
2521 TACTAAAAATAATTTATGCTGGTTCATGAAGATGTACAAAAAAATATTTTTCAAATGGCTAAGGATGGTTTTGATGAAATTGTTACACTAAGGGTCAGCAGTTAA
2626 CAAACCCCATGAAAAAAAAAAAAAAAAAAAAAAAAA
```

**Figure 5  Nucleotide sequence and deduced amino acid sequence of PaPKA.** Initiation codon (ATG) and termination codon (TGA) are highlighted in red boxes; conservative phosphorylation site, DFG triplet and APE motif are highlighted in green boxes; the glycine-rich loop GTGSFGRV (50–57aa), Ser/Thr active site RDLKPEN (165–171aa), PKA-regulatory-subunit-binding site LCGTPEY (198–204aa) are underlined with red.

isoelectric point was 8.35. The nucleotide sequence was deposited in GenBank under accession number KX839259. A glycine-rich loop GTGSFGRV (amino acids 50–57), Ser/Thr active site RDLKPEN (amino acids 165–171), PKA-regulatory-subunit-binding site LCGTPEY (amino acids 198–204), DFG triplet (Asp–Phe–Gly) for orienting the γ-phosphates of adenosine triphosphate (ATP) for transfer, APE motif (Ala-Pro-Glu) to stabilize the structure of the large lobe of PKA, and conserved phosphorylation site (Thr197) were detected in this deduced amino acid sequence. The presence of these conserved regions indicated that *PaPKA* was the catalytic subunit of PKA. An amino acid comparison indicated that PaPKA was highly similar to other PKA catalytic subunits (Fig. 6).

The predicted location of PaPKA in the cell was predominantly in the cytoplasm (47.8%). The three-dimensional structural analysis of PaPKA showed that it folded into a two-lobed structure (Fig. 7). The small lobe had a predominantly β-sheet structure, which was responsible for anchoring and orienting the nucleotide, and the large lobe had a predominantly α-helix structure, and was primarily involved in binding the peptide substrate and initiating phosphotransfer (*Hanks & Hunter, 1995*). Ser53, Phe54, and Gly55 formed hydrogen bonds with ATP β-phosphate oxygens, and Leu49 and Val57 formed a hydrophobic pocket enclosing the adenine ring of ATP.

**Figure 6  Multiple alignment of PaPKA with other PKA.** Amino acid residues that are conserved in at least of 50% sequence are shaded and similar amino acids are shaded in dark. The GenBank accession number for these proteins are as follows: *Aplysia califormica* catalytic subunit of PKA, NP_001191420.1; *Xenopus tropicalis* cAMP dependent protein kinase catalytic subunit, NP_001164667.1; *Branchiostoma floridae* cAMP depedent protein kinase, XP_002600447.1; *Danio rerio* cAMP depedent protein kinase catalytic subunit, NP_001030148.1; *Octopus bimaculoides* cAMP depedent protein kinase catalytic subunit, XP_014777153.1; *Lingula anatina* cAMP depedent protein kinase catalytic subunit, XP_013409439.1; *Crassostrea gigas* cAMP depedent protein kinase catalytic subunit, XP_011439335.1; *Biomphalaria glabrata* cAMP depedent protein kinase catalytic subunit, XP_013072294.1; *Salmo salar* cAMP depedent protein kinase catalytic subunit, XP_014071121.1; *Gallus gallus* cAMP depedent protein kinase catalytic subunit, XP_015146370.1.

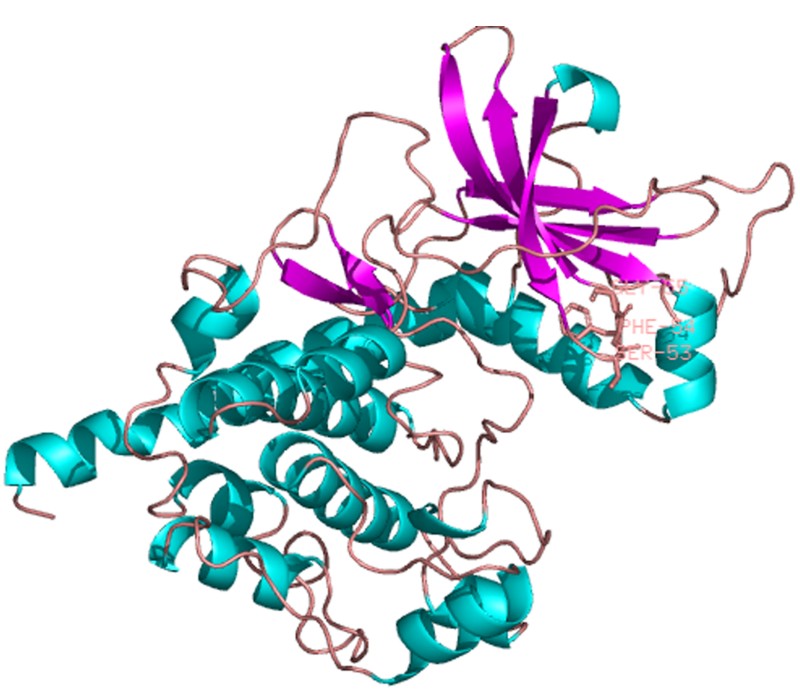

**Figure 7 The three dimensional structure of PKA from *P. aibuhitensis*.** The helix is colored by blue, the sheet is colored by magenta, and the loop is colored by salmon, DFG triplet is labeled in magenta.

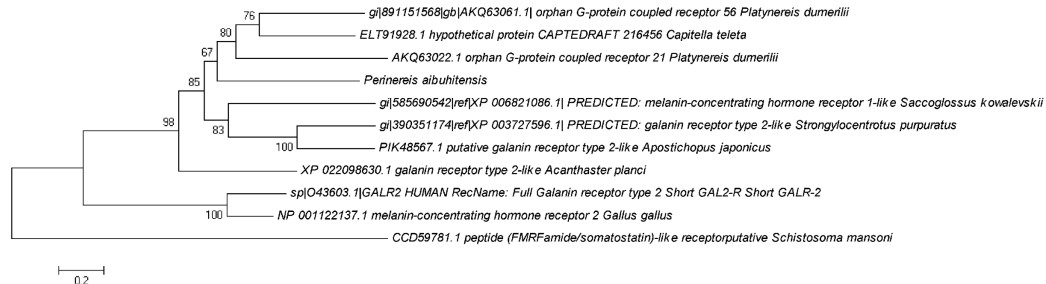

**Figure 8 Phylogenetic analysis of PaGPCR related to GPCR of other invertebrates and vertebrates.** The information of other GPCR are same as the information in Fig. 2; the tree topologies were evaluated with 1,000 replicates.

## Phylogenetic analysis of PaGPCR and PaPKA

Phylogenetic trees were constructed from the amino acid sequences of GPCR and PKA (Figs. 8 and 9, respectively). Figure 8 indicates that PaGPCR shared great identity with the orphan GPCRs of other polychaetes. Figure 9 shows that PaPKA shared identity with mollusk PKAs, which clustered together on a single branch.

## Effects of B(a)P on *PaGPCR* and *PaPKA* expression in *P. aibuhitensis*

Figure 10A shows the expression of the *PaGPCR* gene of *P. aibuhitensis* during B(a)P exposure. There was no difference in its expression between the acetone control group and the blank control group, indicating that acetone as a solvent had no toxic effect on the nematodes. During exposure to B(a)P, the expression of the *PaGPCR* gene increased both

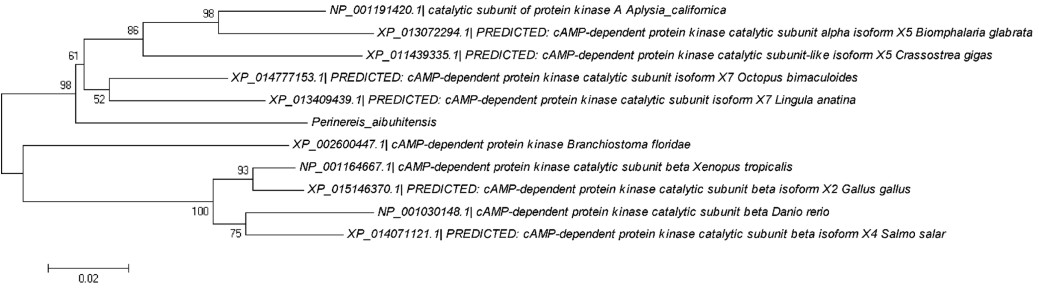

**Figure 9 Phylogenetic analysis of PaPKA related to PKA of other invertebrates and vertebrates.** The information of other PKA sequence are as the information in Fig. 6; the tree topologies were evaluated with 1,000 replicates.

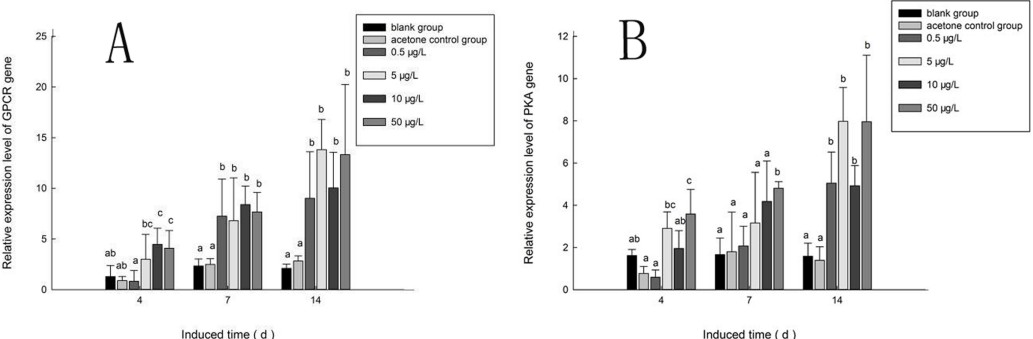

**Figure 10 The relative expression level of *PaGPCR* and *PaPKA* cDNAs under various B(a)P concentration exposure.** (A) represents PaGPCR, (B) represents PaPKA. Different lowercase letters indicate significant difference ($P < 0.05$). all data as mean + SD. $N$ = four worms.

time- and approximately dose-dependently. On day 4, *PaGPCR* expression was significantly upregulated ($P < 0.05$) in all but the 0.5 μg/L B(a)P group. The expression of *PaGPCR* in the 5, 10, and 50 μg/L B(a)P groups was 2.32-, 3.46-, and 3.15-fold higher than in the blank control group, respectively. On day 7, the expression of *PaGPCR* in the 0.5, 5, 10, and 50 μg/L B(a)P groups was 3.10-, 2.91-, 3.59-, and 3.28-fold higher than in the blank control group, respectively ($P < 0.05$). The expression of *PaGPCR* in each concentration group reached its highest level on day 14, at 4.30-, 6.60-, 4.79-, and 6.36-fold higher than the blank control group, respectively ($P < 0.01$).

The expression pattern of the *PaPKA* gene during B(a)P exposure was the same as that of *PaGPCR* (Fig. 10B). The expression of *PaPKA* increased as the time of exposure increased. On day 4, the expression of *PaPKA* was slightly higher in all but the 0.5 μg/L B(a)P concentration group, at 1.79-, 1.21-, and 2.21-fold higher in the 5, 10, and 50 μg/L B(a)P groups, respectively, than in the blank control group. The expression of *PaPKA* in each concentration group was higher on day 7 than on day 4, at 1.25-, 1.90-, 2.52-, and 2.89-fold higher in the 0.5, 5, 10, and 50 μg/L B(a)P groups, respectively, than in the blank control group ($P < 0.05$). On day 14, the expression of *PaPKA* reached its highest level in each concentration group, at 3.19-, 5.03-, 3.10-, and 5.02-fold higher than the blank control group, respectively ($P < 0.05$).

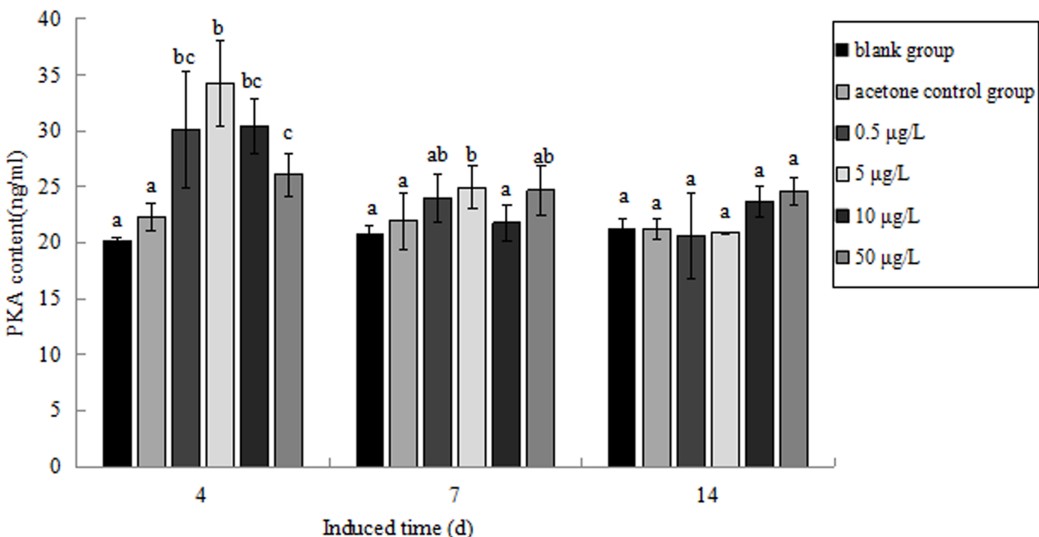

**Figure 11 The PKA content under various B(a)P concentration exposure.** Different lowercase letters indicate significant difference ($P < 0.05$). All data as mean + SD. $N$ = three worms.

## Effect of B(a)P on PKA content in *P. aibuhitensis*

The PKA content in *P. aibuhitensis* under B(a)P exposure was detected (Fig. 11). The PKA content in each concentration group increased and reached its highest level on day 4. The PKA content in each concentration group was 30.09, 34.26, 30.37 and 26.11 ng/mL, respectively, and the content of PKA in 5 µg/L B(a)P concentration group was significantly higher than in the blank control group ($P < 0.05$). On day 7, the PKA content in each concentration group was slightly higher than in the blank control group, but was downregulated than on day 4. The PKA content in the 0.5, 5, 10, and 50 µg/L B(a)P groups was 24.01, 24.97, 21.79 and 24.69 ng/mL, respectively. On day 14, the PKA content in the 0.5, 5 and 50 µg/L B(a)P groups was still downregulated than on day 7, but the PKA content in 10 µg/L B(a)P group was slightly upregulated. The PKA content in each concentration group was 20.62, 20.90, 23.67 and 24.59 ng/mL, respectively.

## DISCUSSION

To investigate the relationship between GPCR signal transduction pathway and B(a)P exposure in *P. aibuhitensis*, the full-length cDNAs of the *PaGPCR* and *PaPKA* were isolated and characterized in *P. aibuhitensis* for the first time. The sequence of *PaGPCR* contained 1,514 bp, encoding 338 amino acids. The deduced protein sequence of PaGPCR contained a 7TM helix bundle domain, flanked by the extracellular N-terminal region and the intracellular C-terminus. As part of the functional mechanism of GPCR, the E/DRY motif (amino acids 122–124), which plays an important role in regulating the conformational state of GPCR, occurred at the border between TM III and intracellular loop 2 in this sequence. The protein sequence of PaGPCR also contained the NPXXY motif (amino acids 298–302) in TM VII, which confirmed that it belonged to the rhodopsin family, the largest of the five families involved in many signaling processes
(*Fredriksson et al., 2003*). A multiple protein sequence alignment showed that PaGPCR shared almost 33% homology with the galanin receptor of echinoderms. A phylogenetic analysis showed that it clustered most closely with the orphan GPCRs of other polychaetes, and the galanin receptor of echinoderms. It is well-known that proteins with similar sequences often display comparable functions if the sequence identify exceeds 30% (*Kakarala & Jamil, 2014*). However, the short transmembrane sequences showed relative low e-values with other GPCRs. If the e-values are low, a prediction based on sequence identity and three dimension structural analysis may not be reliable. Therefore, further study for investigating ligand receptor binding is needed to prove the function of PaGPCR. In contrast to the low sequence identify of GPCR, PKA in *P. aibuhitensis* shared high sequence identify with the PKA catalytic subunits of other species. *PaPKA* contained 2,662 bp, which encoded 350 amino acids. The deduced amino acid sequence of PaPKA contained all the conserved domains that were necessary for kinase activity, such as the conserved Thr in the activation loop, the ATP-binding site (GTGSFGRV), the serine/threonine kinase active site (RDLKPEN), and the PKA-regulatory-subunit-binding site (LCGTPEY). The highly conserved amino acids at the ATP-binding site played important roles in ATP binding and phosphotransfer. The high homology among the PKA catalytic subunits suggested that they have a conserved role in intracellular signaling in both vertebrates and invertebrates.

G-protein-coupled receptors comprise the largest and most important family of cell-surface proteins, transmitting signals from extracellular ligands. In vertebrates, they are key regulators of the innate and adaptive immune responses and have been used as potential targets in drug discovery (*Garland, 2013*). However, they have been inadequately investigated in invertebrates. *Miller et al. (2015)* reported that FSHR-1 mutants of *C. elegans* died significantly more quickly during cadmium exposure than wild-type nematodes, which suggests that the GPCR pathway protects *C. elegans* against pollutant damage. *Dong & Zhang (2012)* reported that Gram-negative bacterial infection induced the expression of the *HP1R* gene in *P. clarkia*. Those results in invertebrates indicate GPCR may also play important role in immune response to environment stimulation. In the present study, we found that B(a)P exposure induced the expression of *PaGPCR*, which increased with time in each concentration group. This result implied that GPCR may play an important role in reducing the deleterious effects of B(a)P in *P. aibuhitensis*. GPCRs interact with diverse chemical structures, which increases cAMP production, which then stimulates phospholipase C activity and the subsequent mobilization of $Ca^{2+}$. *Mayati et al. (2012)* observed an interaction between B(a)P and $\beta_2$ADR in endothelial HMEC-1 cells, which altered the levels of intracellular $Ca^{2+}$ and the expression of cytochrome P450 B1. *Factor et al. (2011)* observed the reduced expression and function of $\beta_2$ADR in airway epithelial cells and smooth muscle cells after their exposure to a mixture of PAHs. In the present study, we observed that the expression of *PaPKA* in *P. aibuhitensis*, was higher during B(a)P exposure than the control level in all but the 0.5 μg/L B(a)P concentration group. The expression of *PaPKA* was significantly and exposure-time-dependently induced by 50 μg/L B(a)P. Besides the gene expression of *PaPKA*, the PKA content was also detected in this study, the PKA content in

*P. aibuhitensis* under B(a)P exposure was upregulated on day 4. *Kreiling, Stephens & Reinisch (2005)* reported exposure to a mixture of bromoform, chloroform and tetrachloroethylene increased cAMP-dependent protein kinase in *Spislula solidissima* embryos. PKA was implicated in regulation of invertebrate haemocyte activity as well as humoral immune response. The increase of PKA combined with GPCR indicated that GPCR pathway in *P. aibuhitensis* was affected by PAHs. After day 4 the PKA content in B(a)P concentration group still higher than in the blank control group, but was downregulated compared to day 4. However, the gene expression of *PaPKA* was time-dependent, the inconsistency of gene expression and protein content showed that there is no simple linear relationship between transcription and translation levels. We speculated that during short time exposure (4 days), the metabolic detoxification in *P. aibuhitensis* was significantly upregulated in order to reduce the toxic effect of PAHs. With time prolonging the organism tends to homeostasis, so the PKA content was not increased continuously.

Benzo(a)pyrene can be metabolized by organisms through a series of enzymatic and nonenzymatic reactions. Typically, B(a)P is metabolized by the phase I enzyme cytochrome P450 (CYP) and phase II enzymes such as glutathione S-transferase. The induction of CYP gene expression has been detected in *P. aibuhitensis* (*Chen et al., 2012*; *Zhao et al., 2014*). In vertebrates, the induction of the CYP enzymes involved in the biotransformation of PAHs is mediated by the aryl hydrocarbon receptor (AhR) pathway. Both α- and β-type AhR proteins have been reported in bivalves (*Fabbri & Capuzzo, 2010*). However, no AhR homologues have been identified in other invertebrates, including marine polychaetes (*Jorgensen et al., 2008*). *Le Ferrec & Øvrevik (2018)* reported that a number of GPCR ligands contain aromatic structures, and that B(a)P modulates the concentration of intracytosolic cAMP through the GPCR pathway without the involvement of conventional nuclear receptors. In this study, we demonstrated the induction of *GPCR* and *PKA* expression during B(a)P exposure, so we hypothesized that the GPCR pathway is also involved in the biotransformation of PAHs in *P. aibuhitensis*. Further study of the relationship between GPCR and CYP expression is required to test our hypothesis.

## CONCLUSIONS

G-protein-coupled receptor represents a critical point of contact between cells and their surrounding environments. This is the first study in which *P. aibuhitensis* GPCR and PKA cDNAs have been cloned. We have also demonstrated that the expression of *GPCR* and *PKA* was induced in *P. aibuhitensis* by B(a)P exposure, and that their expression was affected, to some extent, by the B(a)P concentration and the exposure time. These results should be useful in investigating the biotransformation of PAHs by marine polychaetes.

### Funding

This work was funded by the National Natural Science Foundation of China (No. 41306138), the Key Laboratory of Mariculture and Stock Enhancement in North

China's Sea, Ministry of Agriculture and Rural Affairs, P. R. China (2018-KF-15) and the Liaoning Scientific Instrument Sharing Platform (L201810). The funders had no role in study design, data collection and analysis, decision to publish, or preparation of the manuscript.

## Grant Disclosures

The following grant information was disclosed by the authors:
National Natural Science Foundation of China: 41306138.
Key Laboratory of Mariculture and Stock Enhancement in North China's Sea, Ministry of Agriculture and Rural Affairs, P. R. China: 2018-KF-15.
Liaoning Scientific Instrument Sharing Platform: L201810.

## Competing Interests

The authors declare that they have no competing interests.

## Author Contributions

- Yi Huang conceived and designed the experiments, prepared figures and/or tables, approved the final draft.
- Jia Sun performed the experiments, prepared figures and/or tables, approved the final draft.
- Ping Han analyzed the data, prepared figures and/or tables, approved the final draft.
- Heling Zhao conceived and designed the experiments, contributed reagents/materials/analysis tools, authored or reviewed drafts of the paper, approved the final draft.
- Mengting Wang analyzed the data, prepared figures and/or tables, approved the final draft.
- Yibing Zhou conceived and designed the experiments, authored or reviewed drafts of the paper, approved the final draft.
- Dazuo Yang conceived and designed the experiments, authored or reviewed drafts of the paper, approved the final draft.
- Huan Zhao conceived and designed the experiments, authored or reviewed drafts of the paper, approved the final draft.

## Field Study Permissions

The following information was supplied relating to field study approvals (i.e., approving body and any reference numbers):

The samples in this study were collected from the Dalian Dongyuan aquaculture farm, which is located in the estuary area of Jinzhou Bay in Dalian, China. Our University, Dalian Ocean University, has a long term cooperation agreement with the farm.
The agreement permits us to collect research samples from all their aquaculture sites including a certain estuary area under their ownership.

## Data Availability

Raw data is available in the primary figures and tables and in the Supplemental Files.
## Supplemental Information

Supplemental information for this article can be found online at http://dx.doi.org/10.7717/peerj.8044#supplemental-information.

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
