# Peer review of "Molecular characterization of G-protein-coupled receptor (GPCR) and protein kinase A (PKA) cDNA in Perinereis aibuhitensis and expression during benzo(a)pyrene exposure"

_PeerJ, doi:10.7717/peerj.8044_

## Round 0.1 · original submission · Major Revisions

All the critiques raised by tree reviewers should be taken into account, addressed, and used for the manuscript revision.

Reviewer 1 ·

Basic reporting

The manuscript shows clear and professional organization and sufficient reference and background. All the data were analyzed and interpreted appropriately. The figures and tables are shown with good quality and support the hypotheses.

Experimental design

Methods described in the manuscript is sufficient to follow. Research question well defined, relevant and meaningful.

Validity of the findings

After systematic evaluation, the authors successfully finished molecular characterization of G-protein-coupled receptor (GPCR) and protein kinase A (PKA) cDNA in Perinereis aibuhitensis and expression during benzo(a)pyrene exposure. This new study is absolutely suitable for publication. Data was interpreted completely and supported the conclusion.

Additional comments

Overall, the manuscript contains sufficient data to support its conclusion. The authors investigated the effects of benzo(a)pyrene (B[a]p) on GPCR signaling by using the expression patterns of these two genes during B(a)p exposure determined by real-time fluorescence quantitative PCR. The full-length cDNAs of the PaGPCR and PaPKA were isolated and characterized in P. aibuhitensis for the first time. These results suggested that GPCR signaling in P. aibuhitensis was involved in the polychaete’s response to environmental contaminants. The organization of the full text is scientific and professional for publication.

Reviewer 2 ·

Basic reporting

In this paper, the author has isolated the cDNAs GPCR and PKA in P. aibuhitensis, and also investigated the expression pattern of these two genes during B(a)p exposure. The paper has brought out an interesting hypothesis, however, major changes and supplementary data are needed to improve the quality and integrity of the paper.

Experimental design

1. In Line 114, the author mentioned “four individuals were randomly sampled from each concentration group”, however, in the later section there was only one GPCR sequence generated, were the sequences all the same for the 4 samples?
2. In the structure section, the structures of PaGPCR and PaPKA were only predicted from the amino acid sequences, which is not quite reliable. Further structural study will be needed to fully investigate the structure details of PaGPCR and PaPKA
3. Study on the expression pattern of GPCR gene is not enough to clarify whether GPCR were involved in modulating the toxicity of B(a)p, experiments need to be done using the real proteins, not just the DNAs.

Validity of the findings

See below

Additional comments

1. The paper needs to be examined thoroughly by a native English speaker since there are several gramma mistakes. For example Line 80 “Because of the aromatic structures present a number of GPCR ligands”
2. In Line 283 the author mentioned “This result was similar to reports in other invertebrates”, where are those reports? The author needs to at least cite the references, or generate a new table to compare the data
3. Some of the figures are lack of legends, detailed description will be needed. Also he resolutions of all the figures must be improved

Reviewer 3 ·

Basic reporting

No comment

Experimental design

No comment

Validity of the findings

No comment

Additional comments

In this manuscript, Yi et al describes the effect of benzo(a)pyrene exposure on the expression of two genes in P. aibuhitensis. They cloned and characterized the full-length G-protein-coupled receptor and also protein kinase A cDNAs. The authors found that B(a)p exposure significantly increased the expression of both GPCR and PKA in a time and concentration dependent manner. The authors hypothesized that the GPCR pathway is involved in the biotransformation of polycyclic aromatic hydrocarbons.
The paper is well-structured and clearly written and the analysis of the data is reasonable. Some minor errors need to be addressed:
1. Please check the typological errors in the manuscript.
2. Please replace figure 4 and 7 with higher resolution images, the current ones are highly pixelated.
3. Supplementary images should be referenced in the main text.

---

## Round 0.2 · accepted · Accept

All critiques of both reviewers were adequately addressed and the manuscript was amended accordingly.

Reviewer 2 ·

Basic reporting

The author has revised the manuscript comprehensively, all the points have been addressed carefully. The manuscript is ready for publication now.

Experimental design

N/A

Validity of the findings

N/A

Additional comments

N/A

Reviewer 3 ·

Basic reporting

no comment

Experimental design

no comment

Validity of the findings

no comment

Additional comments

The authors have addressed all my concerns.